# Cryo-EM structures of HKU2 and SADS-CoV spike glycoproteins provide insights into coronavirus evolution

Jinfang Yu [1], Shuyuan Qiao [1], Runyu Guo[1] & Xinquan Wang [1✉]

Porcine coronavirus SADS-CoV has been identified from suckling piglets with severe diarrhea in southern China in 2017. The SADS-CoV genome shares ~95% identity to that of bat α-coronavirus HKU2, suggesting that SADS-CoV may have emerged from a natural reservoir in bats. Here we report the cryo-EM structures of HKU2 and SADS-CoV spike (S) glycoprotein trimers at 2.38 Å and 2.83 Å resolution, respectively. We systematically compare the domains of HKU2 spike with those of α-, β-, γ-, and δ-coronavirus spikes, showing that the S1 subunit N- and C-terminal domains of HKU2/SADS-CoV are ancestral domains in the evolution of coronavirus spike proteins. The connecting region after the fusion peptide in the S2 subunit of HKU2/SADS-CoV adopts a unique conformation. These results structurally demonstrate a close evolutionary relationship between HKU2/SADS-CoV and β-coronavirus spikes and provide insights into the evolution and cross-species transmission of coronaviruses.

[1] The Ministry of Education Key Laboratory of Protein Science, Beijing Advanced Innovation Center for Structural Biology, Beijing Frontier Research Center for Biological Structure, Collaborative Innovation Center for Biotherapy, School of Life Sciences, Tsinghua University, 100084 Beijing, China. ✉email: xinquanwang@tsinghua.edu.cn

Coronaviruses, categorized into the order *Nidovirales*, family *Coronaviridae*, and subfamily *Coronavirinae*, are a large group of viral pathogens with a wide host range[1]. Their infections in humans, other mammals, and birds can cause respiratory, hepatic, enteric and neurological diseases with varying severity[2]. The zoonotic transmission of novel coronaviruses into humans poses severe threats to the global health, evidenced by the severe acute respiratory syndrome coronavirus (SARS-CoV), Middle East respiratory syndrome coronavirus (MERS-CoV), and the SARS-CoV-2 that is causing the ongoing pandemic COVID-19 in the world[1,3–6]. In the meantime, coronaviruses infecting domestic animals also bring substantial economic losses[7]. For example, the swine acute diarrhea syndrome coronavirus (SADS-CoV) (also known as SeACoV and PEAV) isolated in 2017 caused outbreaks of severe watery diarrhea of suckling piglets with a mortality up to 90% in several commercial pig farms in Guangdong Province of China[8–13]. SADS-CoV is an α-coronavirus and other representative members in the α-genus are porcine epidemic diarrhea virus (PEDV), porcine transmissible gastroenteritis coronavirus (TGEV), porcine respiratory coronavirus (PRCV), feline infectious peritonitis virus (FIPV), and human NL63 and 229E coronaviruses (HCoV-NL63 and HCoV-229E)[1]. Representative members in other three genera include mouse hepatitis coronavirus (MHV), bovine coronavirus (BCoV), SARS-CoV, SARS-CoV-2, MERS-CoV, HCoV-OC43, and HCoV-HKU1 in the β-genus, avian infectious bronchitis virus (IBV) in the γ-genus, and porcine deltacoronavirus (PdCoV) in the δ-genus[1].

Cross-species transmission promoted by genetic recombination and/or mutations underlies the host range expansion of coronaviruses[3,14–16]. Bats are the natural reservoir of more than 30 different α- and β-coronaviruses that have great potential for interspecies transmission by recombination and/or mutation[15,17–19]. Data on genetic evolution, receptor binding, and pathogenesis have demonstrated that human SARS-CoV and MERS-CoV most likely originate from bats[1]. Palm civets and dromedary camels are the intermediate hosts of SARS-CoV and MERS-CoV from bats to humans, respectively[1]. The newly identified porcine SADS-CoV isolates are also found to share ~95% sequence identity with *Rhinolophus* bat coronavirus HKU2, and this further stressed the severe results of coronavirus spillover from bats to domestic animals[8–13]. However, the molecular mechanisms underlying the transmission of SADS-CoV from bats to pigs are still unknown and need to be further explored. Recently it was shown that SADS-CoV is able to infect cells from a broad range of species including mouse, chicken, pig, monkey, and human, indicating a high potential of the SADS-CoV for interspecies transmission[20].

The spike glycoprotein of coronaviruses mediates viral entry by binding host receptor with the S1 subunit and fusing viral and cellular membranes with the S2 subunit, thereby determining viral host range and tissue tropism[21,22]. As a class I viral fusion protein, the spike exists on the envelope of virion as a homotrimer and each monomer contains more than 1000 amino acid residues that can be cleaved into S1 and S2 subunits[21]. For most coronaviruses, the N-terminal domain (NTD) of the S1 subunit recognizes cell-surface carbohydrates, while the C-terminal domain (CTD) specifically binds to cellular protein receptors[21–23]. SARS-CoV, SARS-CoV-2, and HCoV-NL63 utilize CTD to bind human receptor ACE2[24–30]; MERS-CoV utilizes CTD to bind human receptor DPP4[31,32]; and TGEV, PRCV, and 229E utilize CTD to bind receptor APN[33–35]; HCoV-OC43 utilizes NTD to recognize glycans[36]; and one exception is MHV, which utilizes the NTD to bind mouse receptor CEACAM1a[37,38]. Therefore, the S1 subunit, especially its NTD and CTD, is the most variable region of the spike, and is responsible for different tropisms of

coronaviruses. In comparison, the S2 subunit containing the fusion peptide (FP) and heptad repeats (HR1 and HR2) for membrane fusion are more conserved in both sequence and structure[21,22]. For the SADS-CoV, receptor analysis indicated that none of the known coronavirus protein receptors including ACE2, DPP4, and APN are essential for cell entry[10,20]. There are also no reports regarding the recognition of glycans by the NTD of SADS-CoV.

Structural studies of the spike and its binding with glycans and protein receptors have provided important insights into the origin, evolution, and interspecies transmission of coronaviruses. Cryo-EM structures of spike trimer from all four coronavirus genera have been reported: the α-coronavirus spike structures are determined for HCoV-NL63[39], HCoV-229E[34], PEDV[40], and FIPV[41]; the β-coronavirus spike structures are determined for MHV[38,42,43], HCoV-HKU1[44], HCoV-OC43[36], SARS-CoV[24,45–47], MERS-CoV[45,48,49], and SARS-CoV-2[50,51]; and the γ-coronavirus spike structure is determined for IBV[52] and the δ-coronavirus spike structure is determined for PdCoV[53,54]. The cryo-EM structures of bat coronavirus spike trimers have not been reported, and only crystal structures of the CTD from HKU4[55], HKU5[56], and HKU9[57] were determined.

The spikes of SADS-CoV (1130 amino acid residues) and HKU2 (1128 amino acid residues) are among the shortest coronavirus spike glycoproteins[58] and their amino acid identities to other known coronavirus spikes are lower than 28%, indicating the spikes of HKU2 and SADS-CoV are unique[8–13,59]. In this study, we report the cryo-EM structures of the SADS-CoV and HKU2 spike trimers at 2.83 and 2.38 Å resolution, respectively. The HKU2 spike trimer structure is the first one from bat coronavirus. We analyze the HKU2 and SADS-CoV trimer structures and also compare the NTD, CTD, SD1, and SD2 domains of the S1 subunit and the S2 subunit of HKU2 with other spikes from α-, β-, γ-, and δ-coronaviruses. Our results strongly support that HKU2 and SADS-CoV preserve primitive structural features in their spikes that have a close evolutionary relationship with β-coronavirus spikes and provide insights into the evolution and cross-species transmission of coronaviruses.

## Results

**Protein expression and structure determination.** The cDNAs encoding HKU2 spike (YP_001552236) and SADS-CoV spike (AVM41569.1) were synthesized with codons optimized for insect cell expression. HKU2 ectodomains (residues 1–1066) and SADS-CoV ectodomains (residues 1–1068) were separately cloned into pFastBac-Dual vector (Invitrogen) with C-terminal foldon tag and Strep tag. After expression in Hi5 insect cells and purification to homogeneity, the cryo-EM images on these two spike ectodomains were recorded using FEI Titan Krios microscope operating at 300 kV with a Gatan K2 Summit direct electron detector (Supplementary Fig. 1). About 1,400,000 particles for HKU2 spike and 900,000 particles for SADS-CoV spike were subjected to 2D classification, and a total of 421,490 particles of HKU2 spike and 152,334 particles of SADS-CoV spike were selected and subjected to 3D refinement with C3 symmetry to generate density maps (Supplementary Fig. 2). The overall density maps were solved to 2.38 Å for HKU2 spike and 2.83 Å for SADS-CoV spike (gold-standard Fourier shell correlation (FSC) = 0.143) (Supplementary Figs. 1–3). The atomic-resolution density map enabled us to build nearly all residues of HKU2 spike ectodomains (residues 17–995) except for a few breaks (residues 129–141 and 203–205), as well as 48 N-linked glycans and 71 water molecules (Supplementary Figs. 3a, 4a, and 5a). The final refined model of SADS-CoV spike contains residues 19–998 with some short breaks (residues 82–83, 134–143, and 488–490) and

**Table 1 Cryo-EM data collection, refinement, and validation statistics.**

|  | HKU2 (EMDB-30037) (PDB 6M15) | SADS-CoV (EMDB-30038) (PDB 6M16) |
|---|---|---|
| Data collection and processing | | |
| Magnification | ×130,000 | ×130,000 |
| Voltage (kV) | 300 | 300 |
| Electron exposure (e−/Å²) | 49.784 | 49.784 |
| Defocus range (μm) | −1 to −3 | −1 to −3 |
| Pixel size (Å) | 1.061 | 1.061 |
| Symmetry imposed | C3 | C3 |
| Initial particle images (no.) | ~1,400,000 | ~900,000 |
| Final particle images (no.) | 421,490 | 152,334 |
| Map resolution (Å) | 2.38 | 2.83 |
| FSC threshold | 0.143 | 0.143 |
| Map resolution range (Å) | 2.38–4 | 2.83–6 |
| Refinement | | |
| Initial model used (PDB code) | 5SZS | 5SZS |
| Model resolution (Å) | 2.38 | 2.83 |
| FSC threshold | 0.143 | 0.143 |
| Model resolution range (Å) | 2.38 | 2.83 |
| Map sharpening B factor (Å²) | −87.05 | −101.51 |
| Model composition | | |
| Nonhydrogen atoms | 23,471 | 23,433 |
| Protein residues | 2895 | 2895 |
| Ligands | 72 | 60 |
| B factors (Å²) | | |
| Protein | 25.63 | 55.82 |
| Ligand | 50.21 | 65.92 |
| R.m.s. deviations | | |
| Bond lengths (Å) | 0.004 | 0.008 |
| Bond angles (°) | 0.624 | 0.763 |
| Validation | | |
| MolProbity score | 1.69 | 2.39 |
| Clashscore | 3.88 | 7.99 |
| Poor rotamers (%) | 3.46 | 3.97 |
| Ramachandran plot | | |
| Favored (%) | 97.5 | 91.68 |
| Allowed (%) | 2.5 | 8.22 |
| Disallowed (%) | 0 | 0.1 |

45 N-linked glycans (Supplementary Figs. 3b, 4b, and 5b). Data collection and refinement statistics for these two structures are listed in Table 1.

**Overall structures of HKU2 and SADS-CoV spikes**. The overall structures of HKU2 and SADS-CoV spikes we determined here resemble the previously reported prefusion structures of coronaviruses spikes. Both spike trimers have a mushroom-like shape (~150 Å in height and ~115 Å in width) (Fig. 1a), consisting of a cap mainly formed by β-sheets of the S1 subunit, a central stalk mainly formed by α-helices of the S2 subunit, and a root formed by twisted β-sheets and loops of the S2 subunit (Fig. 1a). In each trimer there is a C3 axis along the central stalk (Fig. 1a). The amino acid identity between HKU2 and SADS-CoV spikes is 86%, and these two spike structures are quite similar with the root mean square deviation (r.m.s.d.) being 0.53 Å for 962 aligned Cα atoms of the monomer and 0.56 Å for 2886 aligned Cα atoms of the trimer. Due to the high structural similarity, we will use the HKU2 structure to present the features

of both spikes in the subsequent description, whereas significant differences between them will be pointed out only when necessary.

The S1 subunit of the HKU2 spike comprises two major domains, NTD and CTD, which are followed by two subdomains 1 and 2 (SD1 and SD2) connecting them to the S2 subunit (Fig. 1b, c). The S1 subunits from three monomers form the cap of the spike, in which the three CTDs in the inner part are at the apex sitting on top of the central stalk and the three NTDs are located outside the CTDs surrounding the central stalk (Fig. 1a). The NTD, CTD, SD1, and SD2 of the S1 subunit are all mainly composed of β strands (Fig. 1c). In contrast, the upstream helix (UH), FP, connecting region (CR), heptad repeat 1 (HR1), and central helix (CH) of the S2 subunit are mainly composed of helices, whereas the β-hairpin (BH) and subdomain 3 (SD3) at the bottom part of the S2 subunit mainly consist of β strands and loops (Fig. 1c). Moreover, the residues after the SD3, which contain the heptad repeat 2 (HR2), are not resolved in the HKU2 and SADS-CoV spike structures, as well as in all other reported coronavirus spike structures in the prefusion state.

The SD1 and SD2 of the S1 subunit and the S2 subunit are highly similar in amino acid sequence (85, 84, and 95% identities) and structure (Cα r.m.s.d. less than 0.5 Å) between HKU2 and SADS-CoV spikes (Supplementary Fig. 6). The NTD has the lowest sequence identity of 70% among all domains. Structural superimposition also gave a higher Cα r.m.s.d. value of 1.2 Å between these two NTDs. The core β-sheet structure is structurally conserved and conformational variations reside in the loops (Supplementary Fig. 6a). The CTD sequence identity between HKU2 and SADS-CoV spikes is 82%, and the Cα r.m.s.d. between these two CTDs is 1.1 Å, also indicating more structural variations in the CTD in comparison with the highly conserved SD1, SD2, and S2 subunit. The NTD and CTD of the S1 subunit are commonly utilized by coronaviruses for binding cell-surface carbohydrates or protein receptors for cell attachment[23]. Therefore, the sequence and/or structural variations indicate that HKU2 and SADS-CoV would also bind different host receptors by NTD and/or CTD of the S1 subunit, although their receptors in bat and pig are unknown and the receptor-binding sites on spike have not been defined.

**NTD structure and comparisons**. The NTD of HKU2 has three layers of antiparallel β-sheet with the top one consisting of six strands, the middle one consisting of five strands, and the bottom one consisting of three strands. Below the bottom sheet is a short α-helix (Fig. 2a). The top and middle β-sheets form a galectin-like β-sandwich fold, which is inserted between two strands of the bottom sheet (Fig. 2a). To supplement, three disulfide bonds are detected in the HKU2 NTD structure: $C^{17}$–$C^{56}$ connecting the N-terminus of the NTD to its upper loop, $C^{124}$–$C^{149}$ connecting β6 and β7 strands in the top sheet, and $C^{234}$–$C^{244}$ connecting the bottom helix to the bottom sheet (Fig. 2a).

Although the NTDs of all coronaviruses adopt a similar overall architecture, the NTD of HKU2 has the highest structural similarity with the NTD1 (named domain 0 in previous reports) of α-coronavirus HCoV-NL63 with an r.m.s.d. of 2.7 Å for 186 aligned Cα atoms (Fig. 2b and Supplementary Fig. 8d). The NTD1 of α-coronavirus PEDV is not completely modeled in the spike trimer structure; however, the partial model still fits well with the NTD of HKU2 with an r.m.s.d. of 2.3 Å for 73 aligned Cα atoms. Both HCoV-NL63 and PEDV have a second NTD (NTD2, also named domain A in previous reports), and the NTD of HKU2 is structurally less similar to the NTD2 with an r.m.s.d. of 4.3 Å against HCoV-NL63 NTD2 and of 4.1 Å against PEDV NTD2 (Fig. 2b and Supplementary Fig. 8d). Recent structural

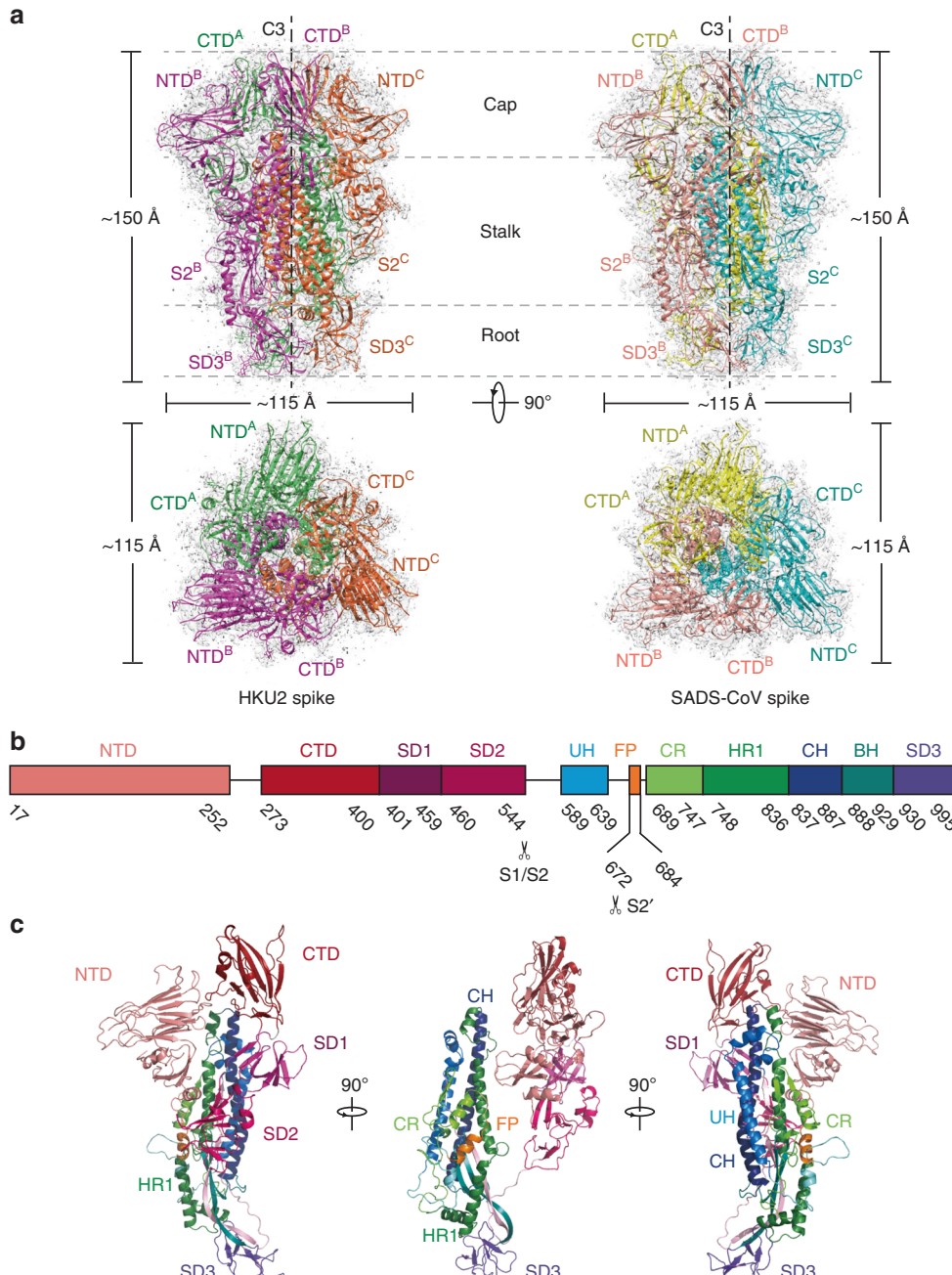

**Fig. 1 Overall structures of HKU2 and SADS-CoV spike glycoproteins. a** Overall structures of HKU2 and SADS-CoV spike glycoproteins shown in side view (upper panel) and top view (lower panel). Three monomers of HKU2 spike are colored magenta, green, and orange, respectively; three monomers of SADS-CoV spike are colored pink, yellow, and cyan, respectively. The cryo-EM maps are shown as semitransparent surface and contoured at 2.6 RMS and 3 RMS for HKU2 and SADS-CoV spikes, respectively. The trigonal axes are shown as black dashed lines. Visible segments of each monomer are labeled accordingly. The cap, stalk, and root parts are partitioned by gray dashed lines. **b** Segmentation of HKU2 monomer. The segments of HKU2 are shown as boxes with the width related to the length of amino acid sequence. The start and end amino acids of each segment are labeled. The position of S1/S2, and S2′ cleavage sites are indicated. NTD N-terminal domain, CTD C-terminal domain, SD1 subdomain 1, SD2 subdomain 2, UH upstream helix, FP fusion peptide, CR connecting region, HR1 heptad repeat 1, CH central helix, BH β-hairpin, SD3 subdomain 3. **c** Overall structure of HKU2 monomer. Side views of HKU2 monomer shown in three directions. The segments are colored the same as in **b**.

determination showed that another α-coronavirus, HCoV-229E, also has one NTD, which is more structurally similar to the NTD2 (Cα r.m.s.d. of 2.0 Å) than the NTD1 (Cα r.m.s.d. of 3.7 Å) of HCoV-NL63 (Fig. 2c). All above comparisons indicate that there are two subtypes of NTD in the α-coronaviruses: the subtype I is represented by the NTDs of HKU2, SADS-CoV, the NTD1s of HCoV-NL63 and PEDV, and the subtype II is represented by the NTD of HCoV-229E, the NTD2s of HCoV-NL63 and PEDV (Fig. 2b, c). Although sharing an overall architecture, these two NTD subtypes have a structural difference in the galectin-like β-sandwich fold containing the top and middle sheets stacked together through hydrophobic interactions. These two β-sheets are well aligned in the galectin-like domain of subtype I, whereas there is an alignment shift in the galectin-like

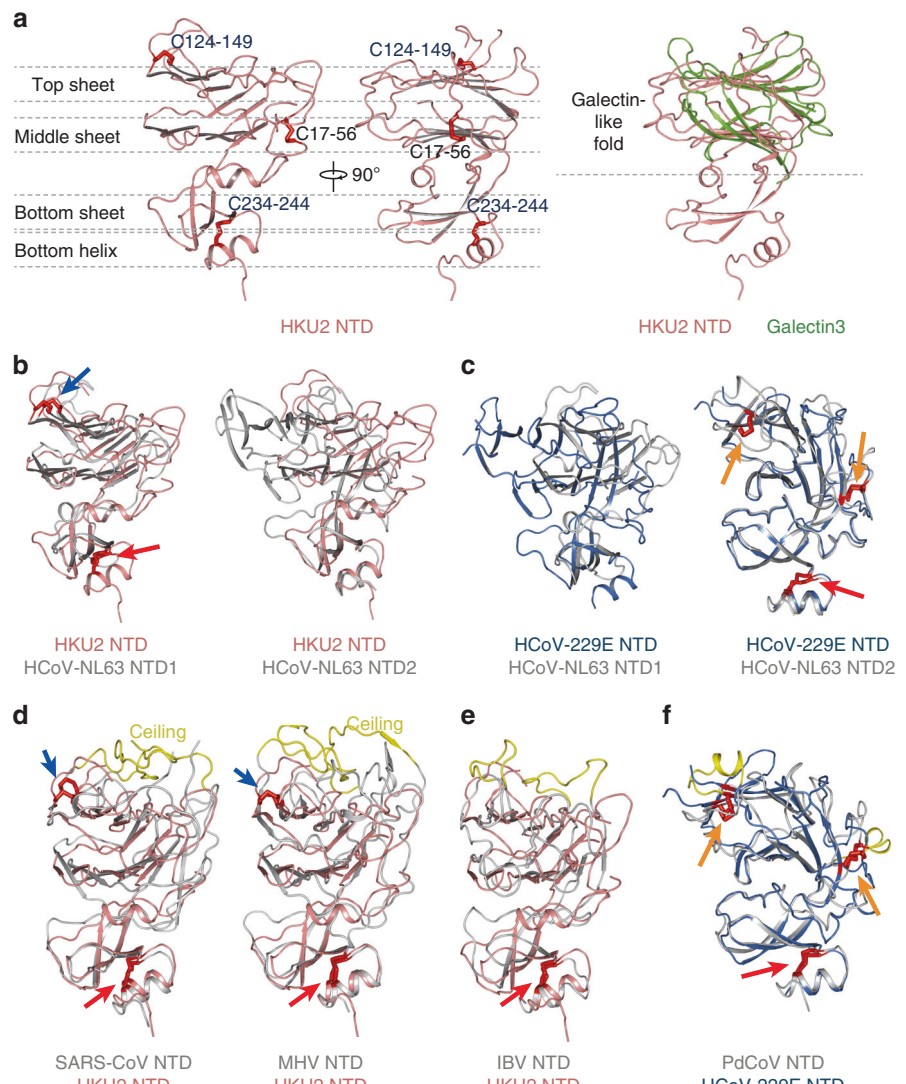

**Fig. 2 Structure of HKU2 NTD and comparisons. a** Structure of HKU2 NTD. Side views of HKU2 NTD are shown in two orthogonal directions. Disulfide bonds are shown as red sticks. Conserved disulfide bonds are labeled blue; other disulfide bonds are labeled black. Top sheet, middle sheet, bottom sheet, and bottom helix are partitioned by gray dashed lines. Comparison of HKU2 NTD and galectin3 are shown in the right panel. HKU2 NTD is colored salmon; galectin3 is colored green. PDB code: galectin3, 1A3K. **b** Subtype I NTD of α-coronavirus. Structural alignments of HKU2 NTD with HCoV-NL63 NTD1, and with HCoV-NL63 NTD2 are shown. HKU2 NTD is colored salmon. HCoV-NL63 NTD1 and NTD2 are colored gray. Disulfide bonds are shown as red sticks. Disulfide bonds conserved in both types of NTDs are indicated by red arrows; disulfide bonds conserved in subtype I NTD are indicated by blue arrows; PDB code: HCoV-NL63, 5SZS. **c** Subtype II NTD of α-coronavirus. Structural alignments of HCoV-229E NTD with HCoV-NL63 NTD1, and with HCoV-NL63 NTD2 are shown. HCoV-229E NTD is colored marine. HCoV-NL63 NTD1 and NTD2 are colored gray. Disulfide bonds conserved in both types of NTDs are indicated by red arrows; disulfide bonds conserved in subtype II NTD are indicated by orange arrows. PDB codes: HCoV-229E, 6U7H. **d** β-coronavirus NTDs resemble subtype I. Structural alignments of HKU2 NTD with SARS-CoV NTD, and with MHV NTD are shown. Disulfide bonds are shown and labeled the same as in **b**. Ceilings in β-coronavirus NTDs are shown in yellow. PDB codes: SARS, 5XLR; MHV, 3JCL. **e** γ-coronavirus IBV NTD resembles subtype I. Structural alignment of HKU2 NTD with IBV NTD is shown. Disulfide bonds are shown and labeled the same as in **b**. The additional loops in IBV NTD is shown in yellow. PDB code: IBV, 6CV0. **f** δ-coronavirus PdCoV NTD resembles subtype II. Structural alignment of HCoV-229E NTD with PdCoV NTD is shown. Disulfide bonds are shown and labeled the same as in **c**. The additional helices in PdCoV NTD are shown in yellow. PDB codes: HCoV-229E, 6U7H; PdCoV, 6B7N.

domain of subtype II (Fig. 2b). The other notable difference is the distribution of signature disulfide bonds. A signature disulfide bond $C^{124}–C^{149}$ (numbered in HKU2 and connecting β6 and β7 strands in the top sheet) is conserved in all subtype I NTDs (Fig. 2b), and the subtype II NTDs have two signature disulfide bonds: the first one $C^{145}–C^{168}$ (numbered in HCoV-229E) connecting β5 and β6 strands and the second one $C^{81}–C^{105}$ (numbered in HCoV-229E) connecting β2 and neighbor loop (Fig. 2c). Besides, $C^{234}–C^{244}$ (numbered in HKU2 and connecting

the bottom helix to the bottom sheet) is conserved in both subtype I and subtype II NTDs (Fig. 2b, c).

The NTDs of β-coronaviruses including BCoV, HCoV-HKU1, HCoV-OC43, MERS-CoV, SARS-CoV, and MHV resemble the subtype I, rather than the subtype II NTD in the topology and distribution of the disulfide bonds (Fig. 2d). These β-coronavirus NTDs have additional loops in the N-terminus, between β1 and β2 strands, and between β6 and β7 strands (numbered in HKU2 structure), forming an extensive ceiling-like structure on

top of the galectin-like fold (Fig. 2d). It has been found that the evolvement of this ceiling-like structure has functional outcomes such as immune evasion or receptor binding[52]. The NTD of γ-coronavirus IBV also resemble the subtype I NTD in the topology, although its disulfide bond positions are not conserved as in the subtype I NTD (Fig. 2e). Of note, the NTD of δ-coronavirus PdCoV resemble the subtype II NTD in the topology and distribution of the disulfide bonds (Fig. 2f). Both IBV and PdCoV NTDs also have additional insertions including loops and short helices in the galectin-like fold compared with the two subtypes of NTD in α-coronaviruses (Fig. 2e, f).

**CTD structure and comparisons.** The CTD of HKU2 has a twisted five-stranded antiparallel β-sheet as the core with connecting loops between the strands (Fig. 3a). It contains four disulfide bonds: $C^{277}$–$C^{300}$ and $C^{285}$–$C^{290}$ at the N-terminus, $C^{341}$–$C^{397}$ at the C-terminus, and the last one $C^{331}$–$C^{369}$ connecting the β2 and β5 strands in the core β-sheet (Fig. 3a). Interestingly, the CTD core of HKU2 is of high structural similarity with the conserved CTD core of β-coronaviruses and the disulfide bonds in the CTD of HKU2 except for $C^{285}$–$C^{290}$ are also detected in all β-coronavirus CTDs (Fig. 3b). These CTDs have the core of one twisted β-sheet and here we name them as one-layer CTD subtype (Fig. 3a, b). The β-coronavirus CTDs always have an insertion consisting of loops and/or strands between the β5 and β6 strands of the core (Fig. 3b). SARS-CoV[26], SARS-CoV-2[27–30], MERS-CoV[31,32], HKU4[55], and HKU5[56] have receptor-binding motif (RBM) in this insertion region responsible for binding their respective protein receptors. In the CTD of HKU2, there is only one short loop between the β5 and β6 strands of the core twisted β-sheet (Fig. 3a).

Although as members in the α-genus, HKU2, and SADS-CoV CTD structures are significantly different from those of other α-coronaviruses HCoV-NL63, HCoV-229E, PEDV, TGEV, and PRCV that contain two layers of β-sheets (Fig. 3c). And we named these CTDs as two-layer CTD subtype. All available two-layer CTD structures can be well aligned with Cα r.m.s.d. in the range of 1.0–3.4 Å. These two-layer CTDs contain two highly conserved disulfide bonds: $C^{540}$–$C^{586}$ and $C^{569}$–$C^{596}$ (numbered in PEDV CTD) (Fig. 3c). The $C^{569}$–$C^{596}$ is conserved among all coronaviruses, whereas the $C^{540}$–$C^{586}$ is conserved in all α-coronaviruses (except for HKU2 and SADS) and δ-coronavirus PdCoV (Fig. 3c).

The CTD of δ-coronavirus PdCoV has a core of two β-sheets, belonging to the two-layer CTD subtype (Fig. 3c). As for the γ-coronavirus IBV, the core of its CTD is also similar to the typical two-layer CTD (Fig. 3c). However, several β strands are replaced by loops and the disulfide bonds are in different positions from the two-layer CTD (Fig. 3c). IBV CTD also has an extra region of loops, reminiscent of the extra domain in the CTDs of β-coronaviruses (Fig. 3c).

**SD1 and SD2 structures and comparisons.** The SD1 and SD2 are two subdomains following the CTD in the S1 subunit, linking the CTD to the S2 subunit. The HKU2 SD1 is a partial β barrel consisting of five β strands and a disulfide bond ($C^{409}$–$C^{458}$) connecting its C-terminus to the β1 strand (Fig. 4a). This five-stranded β barrel and the linking disulfide bond are conserved among all four genera of coronavirus (Fig. 4a). The HKU2 SD2 has a structure of two layers of β-sheet with an additional short α-helix over the top sheet (Fig. 4b). The additional α-helix and the top sheet is linked by a disulfide bond ($C^{482}$–$C^{509}$), and another disulfide bond ($C^{524}$–$C^{533}$) links the C-terminal loop to the bottom sheet (Fig. 4b). The two-layer core structure and the second disulfide bond are conserved among all genera of

coronavirus; however, the additional α-helix and the first linking disulfide bond is a distinct feature of β-coronaviruses plus α-coronavirus HKU2 and SADS-CoV (Fig. 4b). This additional helix appears to be an insertion between the β2 and β3 strands of the SD2, and is retained during evolution of β-coronaviruses.

**Quaternary packing of the NTD and CTD in the spike.** It has been observed that coronaviruses have two types of quaternary packing mode of the S1 subunits in the trimer: intrasubunit packing and cross-subunit packing[52]. Actually, this is mainly caused by different positioning and interaction between NTD and CTD in the spike monomer. The HKU2 S1 subunit, similar to those in α-coronaviruses HCoV-NL63, HCoV-229E, and PEDV and δ-coronavirus PdCoV, has an inward CTD that contacts with the NTD (Fig. 5a). The three structural NTD–CTD modules in the cap region of these spikes are composed of NTD and CTD from the same monomer, forming the intrasubunit packing in the spike trimer (Fig. 5a). The S1 subunits of other coronaviruses in the β- and γ-genera including MHV, SARS-CoV, MERS-CoV, HCoV-OC43, and IBV have an outward CTD that is far away from the NTD (Fig. 5b). Therefore, the three structural NTD–CTD modules in the cap region of these spikes have the NTD from one monomer and the CTD from the adjacent monomer, forming the cross-subunit packing in the spike trimer (Fig. 5b). Interestingly, we found that the outward CTDs always have an insertion in the core structure, such as β-coronavirus CTDs and γ-coronavirus IBV CTD (Fig. 5b). In contrast, all inward CTDs only have the one-layer or two-layer core structure without obvious inserted region.

**Conserved S2 subunit and a distinct CR.** Sequence analysis suggested that the S1/S2 protease cleavage site at the boundary between the S1 and S2 subunits is R544-M545 in HKU2 spike and R546-M547 in SADS-CoV spike[8,11,59]. Compared with the S1 subunit, the topology and structure of S2 subunit are highly conserved in all coronavirus spikes. The HKU2 S2 subunit contains a 3-helix UH (residues 589–639), an FP helix (residues 672–684), a CR (residues 689–747), a 4-helix HR1 (residues 748–836), a CH (residues 837–887), a twisted BH (residues 888–929), and a β-sandwich like SD3 (residues 930–995) (Figs. 1b and 6a). Like in other coronavirus spikes in the prefusion state, the model of HR2 after SD3 was not built in the structure due to poor density. Five disulfide bonds in S2 are detected. Two of them ($C^{590}$–$C^{612}$ and $C^{595}$–$C^{601}$) stabilize the folded helices of UH, $C^{696}$–$C^{706}$ bends the CR, $C^{884}$–$C^{895}$ links the CH and the BH, and $C^{934}$–$C^{943}$ is within the SD3 (Fig. 6a). The first four disulfide bonds are conserved in all coronaviruses, and the last one in the SD3 has different positions in different spikes. Specifically, it links the β2 and β3 strands of SD3 in the spikes of HKU2, SADS-CoV, and MERS-CoV (numbered in MERS-CoV), and in other coronavirus spikes it links the β2 strand to the C-terminal loop of SD3 (numbered in MERS-CoV) (Supplementary Fig. 7).

All coronavirus spikes have the S2′ protease site upstream from the FP in the S2 subunit, which is essential for proteolytic fusion activation of the spike. Receptor binding and cleavage at the S2′ site promote large-scale conformational changes of the FP, CR, HR1, and HR2, allowing the insertion of FP into host cell membrane and the formation of six-helix bundle. The FP and CR, which are often not well and totally resolved in other coronavirus spike structures, can be clearly modeled in the HKU2 spike due to the atomic resolution of the map (Supplementary Fig. 5). The typical CR in the S2 subunit contains three helices and one short strand, with a disulfide bond bending the first and second helix to form a turn (Fig. 6b). In HKU2, the second helix is replaced by a

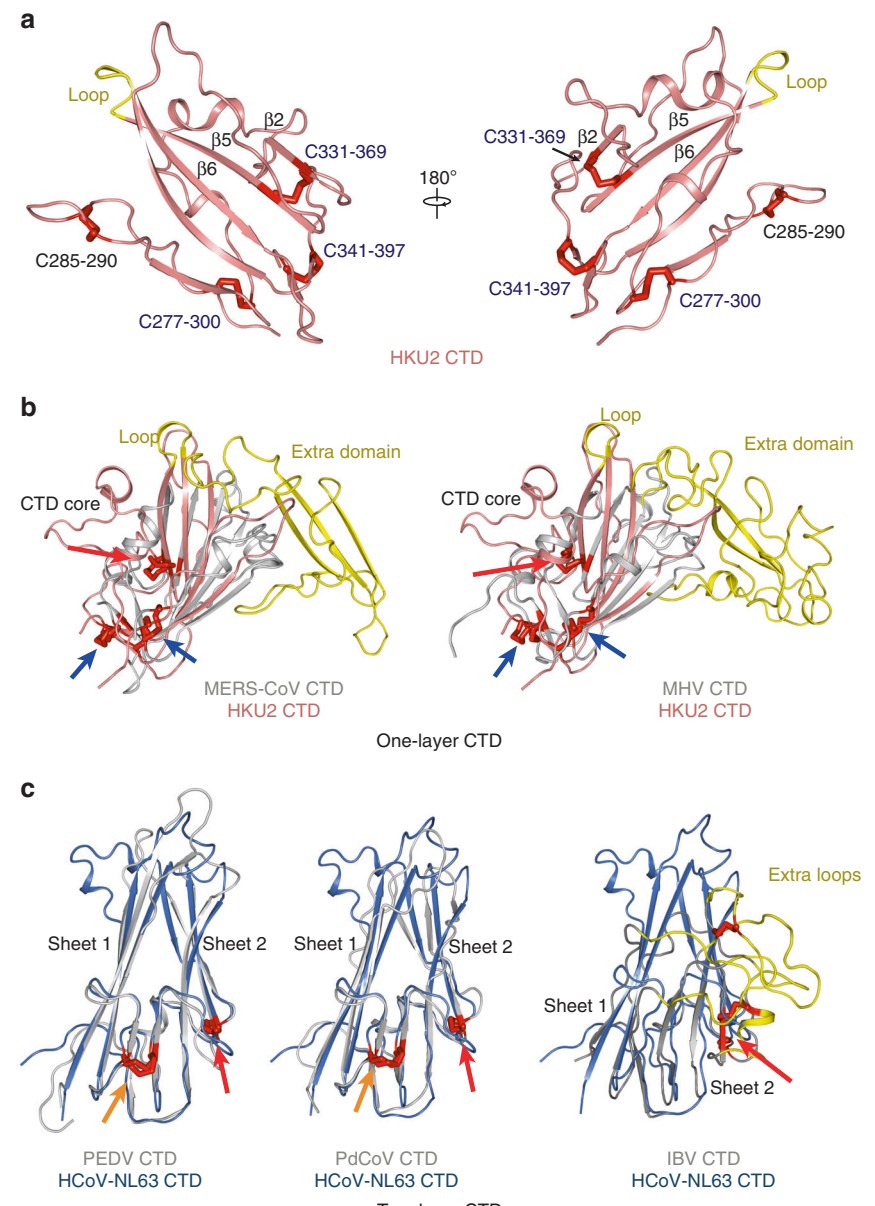

**Fig. 3 Structure of HKU2 CTD and comparisons. a** HKU2 CTD shown in two opposite directions. Strands mentioned in the main text are labeled. The loop replaced by extra domain in β-coronavirus CTDs is shown in yellow. Disulfide bonds are shown as red sticks. Conserved disulfide bonds are labeled blue; other disulfide bonds are labeled black. **b** β-coronavirus CTDs belong to one-layer CTD. Structural alignments of HKU2 CTD with MERS-CoV CTD, and with MHV CTD are shown. Extra domains in β-coronavirus CTDs are colored yellow. Disulfide bonds are shown as red sticks. Disulfide bonds conserved in both one-layer CTD and two-layer CTD are indicated by red arrows. Disulfide bonds only conserved in one-layer CTD are indicated by blue arrows. PDB codes: MHV, 3JCL; MERS-CoV, 6Q05. **c** α-coronavirus (except for HKU2 and SADS-CoV), γ-coronavirus, and δ-coronavirus CTDs belong to two-layer CTD. Structural alignments of HCoV-NL63 CTD with PEDV CTD, with PdCoV CTD, and with IBV CTD are shown. Two layers of β-sheets are labeled. Extra loops in IBV CTD are colored yellow. Disulfide bonds are shown as red sticks. Disulfide bonds conserved in both one-layer CTD and two-layer CTD are indicated by red arrows. Disulfide bonds only conserved in two-layer CTD are indicated by orange arrows. PDB codes: HCoV-NL63, 5SZS; PEDV, 6U7K; IBV, 6CV0; PdCoV, 6B7N.

short strand (713–716) and the third helix is replaced by a loop (721–741), therefore there are two short strands and only one helix in HKU2 CR (Fig. 6c). The conserved disulfide bond $C^{696}$–$C^{706}$ makes the first helix of CR in HKU2 spike turn upside down. The S2′ cleavage site (between R671 and S672) is then covered by the reversed CR helix and loops, and R671 interacts with E723 in the loop and K697 and K698 in helix 1 (Fig. 6c). In other coronaviruses, taking the MHV S2 for example, the helix 1 does not cover the S2′ site (between R869 and S870), and R869 only loosely interacts with T929 in helix 3 (Fig. 6b). After the

dissociation of the S1 subunit triggered by receptor binding, the exposure of the S2′ site for cleavage is a prerequisite for the proteolytic activation of the coronavirus spike to mediate membrane fusion. The buried S2′ site indicates that HKU2 spike, compared with other coronavirus spikes, would require more conformational changes around the S2′ site for the exposure.

## Discussion
A porcine coronavirus SADS-CoV (also named SeACoV and PEAV in other reports) was recently identified from suckling

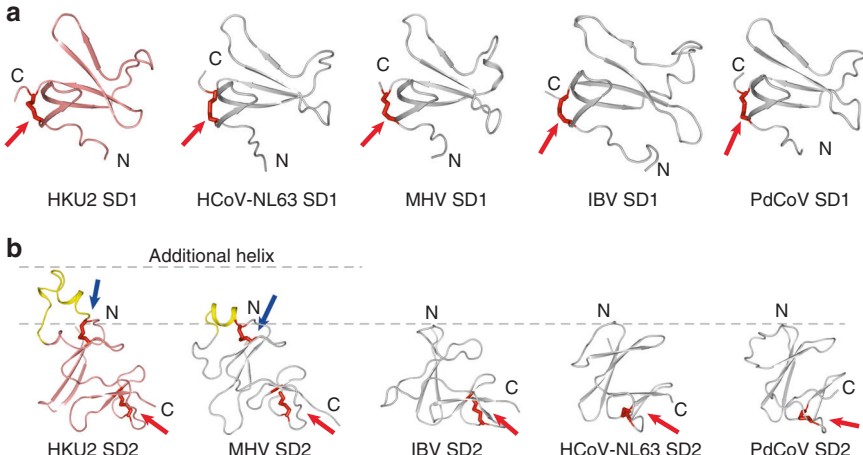

**Fig. 4 Structures of SD1 and SD2 and comparisons. a** Comparison of SD1 from four genera of coronaviruses. HKU2 SD1 is colored salmon; SD1 from other coronaviruses are colored gray. Disulfide bonds are shown as red sticks. Red arrows indicate the disulfide bonds conserved in all genera of coronaviruses. PDB codes: HCoV-NL63, 5SZS; MHV, 3JCL; IBV, 6CV0; PdCoV, 6B7N. **b** Comparison of SD2 from four genera of coronaviruses. HKU2 SD2 is colored salmon; SD2 from other coronaviruses are colored gray. Disulfide bonds are shown as red sticks. Red arrows indicate the disulfide bonds conserved in all genera of coronaviruses. Blue arrows indicate the disulfide bonds only found in HKU2 (and SADS-CoV) and β-CoVs. The additional helices of SD2 from HKU2 (and SADS-CoV) and βCoVs are colored yellow and partitioned by gray dashed lines. PDB codes: HCoV-NL63, 5SZS; MHV, 3JCL; IBV, 6CV0; PdCoV, 6B7N.

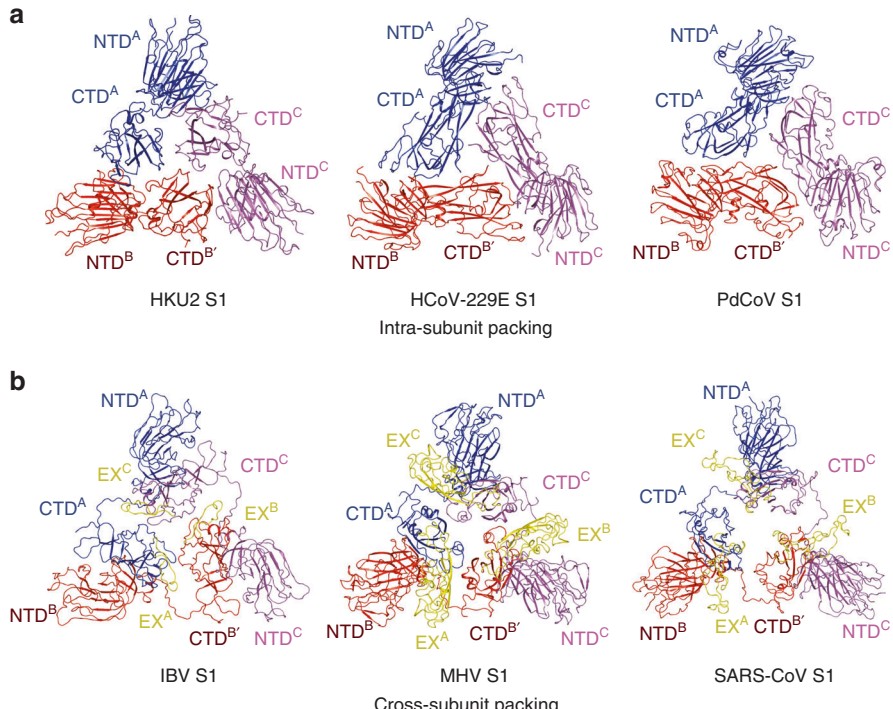

**Fig. 5 Quaternary packing of NTD and CTD. a** α-coronavirus S1 and δ-coronavirus S1 use intrasubunit packing pattern. NTD and CTD from the first monomer are colored blue, the second are colored red, and the third are colored magenta. PDB codes: HCoV-229E, 6U7H; PdCoV, 6B7N. **b** β-coronavirus S1 and γ-coronavirus S1 use cross-subunit packing pattern. NTD and CTD from the first monomer are colored blue, the second are colored red, and the third are colored magenta. The extra loop of IBV and the extra domains of β-CoVs are colored yellow and labeled as EX. PDB codes: MHV, 3JCL; IBV, 6CV0; SARS, 5XLR.

piglets with diarrhea in southern China, and its genome sequence was most identical (~95% identity) to that of *Rhinolophus* bat α-coronavirus HKU2[8–13]. The SADS-CoV and HKU2 are phylogenetically located in a sublineage closely related to the proposed α-coronavirus group-1b lineage at the complete genome level[8–13]. However, phylogenetic analysis based on the spike glycoprotein indicated that they are members of a separate lineage clustered within β-coronavirus (Supplementary Fig. 8a), suggesting that HKU2 and SADS-CoV probably resulted from recombination of an α-coronavirus with an unrecognized β-coronavirus S gene[8–13]. These results, together with the lower than 28% amino acid identities to other known coronavirus glycoproteins, strongly indicate that the spike glycoproteins of HKU2 and SADS-CoV are unique[8–13]. In this study, we determined the cryo-EM structures

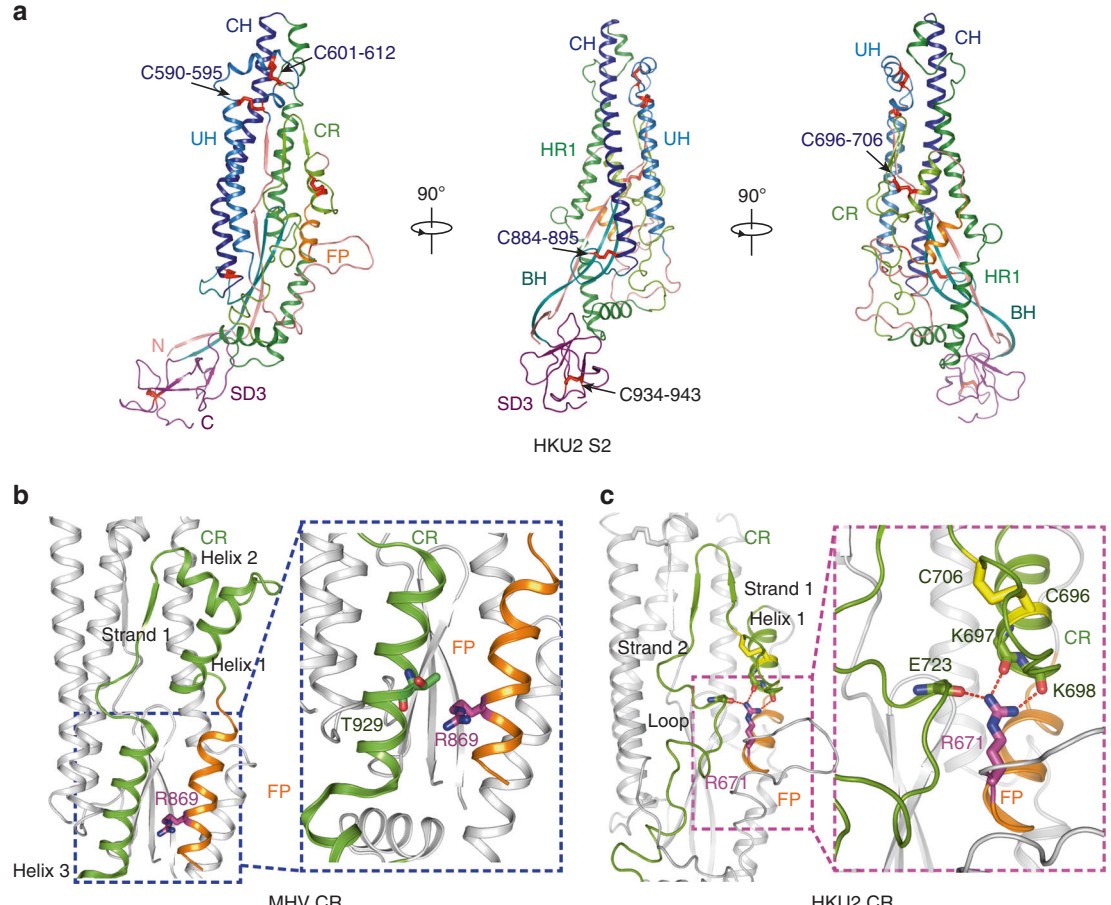

**Fig. 6 Structure of HKU2 S2. a** Side views of HKU2 S2 shown in three directions. Seven segments of S2 are shown as different colors. UH upstream helix, FP fusion peptide, CR connecting region, HR1 heptad repeat 1, CH central helix, BH β-hairpin, SD3 subdomain 3. Disulfide bonds are shown as red sticks. Disulfide bonds conserved in all coronaviruses are labeled blue; the other disulfide bond is labeled black. **b** Conserved CR represented by MHV CR. CR and FP are colored the same as in **a**. Helices and strands in CR are labeled. R869 (S2′ cleavage site) is shown as stick and colored magenta. The blue dashed box shows R869 does not interact tightly with MHV CR. PDB code: MHV, 3JCL. **c** Unique feature of HKU2 CR. CR and FP are colored the same as in **a**. Helices and strands in CR are labeled. R671 (S2′ cleavage site) is shown as stick and colored magenta. The magenta dashed box shows detailed interactions between R671 and CR.

of HKU2 and SADS-CoV spike glycoproteins at atomic resolutions. Pairwise comparisons demonstrated nearly identical overall structures, and differences mainly locate in the loops of NTD and CTD of the S1 subunit between the spikes of HKU2 and SADS-CoV (Supplementary Fig. 6). A series of structural analysis and comparisons were also performed at the domain level between HKU2 spike with other coronavirus spikes with determined structures. Our results show that HKU2 and SADS-CoV spikes maintain primitive structural features, especially in the NTD and CTD, and provide more insights into the evolution of coronaviruses.

The HKU2 and SADS-CoV have one NTD in the S1 subunit, and their structures are more similar to the NTD1 than the NTD2 of α-coronaviruses HCoV-NL63 and PEDV, whereas the only NTD of HCoV-229E is structurally more similar to the NTD2 than the NTD1 (Fig. 2). Therefore, we suggest that α-coronaviruses have two subtypes of NTD. The evolution relationship between them are not clear yet. It was once suggested that the presence of two NTDs in HCoV-NL63 is a result of gene duplication[39]. However, the sequence identity between these two NTDs is only 15.7% in HCoV-NL63 and 12.9% in PEDV. Considering that HKU2 (SADS-CoV) and HCoV-229E have one NTD belonging to either subtype I or subtype II, a more plausible

evolution way of the NTD in α-coronaviruses is the recombination of two separate primitive domains into the genome, resulting in the presence of two NTDs in the S1 subunit of α-coronaviruses including HCoV-NL63 and PEDV. Of note, these two NTD subtypes may represent primitive structures that could be the evolutionary ancestors of NTDs of other genera coronaviruses. For example, in the current available spike structures, the NTDs of β-coronavirus are similar to the HKU2 NTD representing the subtype I in both architecture and disulfide bond positions. These β-coronavirus NTDs also have additional loop ceiling over the top sheet, functionally facilitating immune evasion or protein receptor binding such as in MHV[37,38]. The NTD of γ-coronavirus IBV is also architecturally similar to the subtype I, although the disulfide bond positions are not conserved (Fig. 2e). In contrast, the δ-coronavirus PdCoV NTD is similar to the HCoV-229E NTD representing the subtype II in both architecture and disulfide bond positions (Fig. 2f). A previous study of the IBV spike proposed that α-coronavirus NTDs are probably the most ancestral and the NTDs of the four genera form an evolutionary spectrum in the order of α-, δ-, γ-, and β-genus[52]. Our proposal here is similar to the previous one in the point that two NTD subtypes in α-coronaviruses may represent primitive structures that could be the evolutionary ancestors of NTDs. However, we

argue that the evolution pathways may not be in the order of α-, δ-, γ-, and β-genus. A more plausible pathway is that the β-, γ-, and δ-coronavirus NTDs may evolve independently and parallelly from subtype I (β- and γ-coronavirus NTDs) or subtype II (δ-coronavirus NTDs) (Supplementary Fig. 8b).

The HKU2 and SADS-CoV CTD structure has a one-layer core consisting of a twisted five-stranded antiparallel β-sheet. Interestingly, β-coronavirus CTDs also have the similar one-layer core structure and three strictly conserved disulfide bonds are also present in the core of HKU2 CTD. Currently, all identified RBM of β-coronavirus CTDs are within an inserted domain between two strands of the core sheet, and this insertion responsible for receptor binding of β-coronaviruses is replaced by a short loop in HKU2 CTD. This result firstly indicates HKU2 CTD represent a primitive structure in the one-layer CTD family, while the inserted domain in β-coronaviruses results from a recombinant event during evolution (Supplementary Fig. 8c). The second indication is that HKU2 and SADS-CoV may not utilize the CTD to bind protein receptors that have not been identified yet, and their different receptor usage may be determined by the NTD that harbors almost 50% of residue difference between them. Of note, the CTDs from other α-coronaviruses, γ-coronavirus IBV, and δ-coronavirus PdCoV all belong to the two-layer subtype consisting of two layers β-sheets, although with structural variations in different viruses. These results further confirmed the previous phylogenetic analysis suggesting that HKU2 and SADS-CoV probably resulted from a recombination of an α-coronavirus genomic backbone with an unrecognized β-coronavirus spike gene[8,11,59].

In contrast with the NTD and CTD having unique structural features, the SD1 and SD2 of the S1 subunit and the S2 subunits of HKU2 and SADS-CoV are structurally conserved as those of other coronaviruses. In the evolutionary aspect, it is not surprising because this region either connects the CTD to the S2 subunit (SD1 and SD2) or mediates the membrane fusion (S2 subunits), whereas the NTD and CTD are key factors determining tissue tropism and host range of coronaviruses[23]. Even highly conserved in overall structure, the S2 subunit in HKU2 and SADS-CoV still have a secondary structure arrangement in the CR after the FP, resulting in a more buried S2′ cleavage site (Fig. 6c). It indicates that although the membrane fusion mechanism is highly conserved, the dynamic fusion procedures of HKU2 and SADS-CoV may still have their unique features that need to be addressed in the future.

The ongoing COVID-19 pandemic caused by SARS-CoV-2 has raised worldwide alert[4–6]. We compared the domain structures of HKU2 and SARS-CoV-2 S proteins (Supplementary Fig. 9). The NTD of SARS-CoV-2 has the two conserved disulfide bonds in subtype I; the CTD core of SARS-CoV-2 has one layer of β-sheets; the SD1 and SD2 of SARS-CoV-2 have the same disulfide bonds with HKU2, although the additional helix of SD2 is not built in the SARS-CoV-2 S model owing to poor density in this region. SARS-CoV-2 has the typical β-coronavirus structural features in each domain, and the inserted RBM in the CTD could be the major impetus for cross-species transmission from bats to human.

In comparing the structures of HKU2 and SADS-CoV spikes with other coronavirus spikes, we observed that additional NTDs could be recombined, ceiling loops could be inserted into NTD core, and extra domain containing RBM could be inserted into the CTD core structure. These phenomena indicate the subdomains are gradually recruited into the S1 subunit during evolution, and the recruitments are required for cross-species transmission, adapting to different host range, and responding to the updating of host immune system, which provides a vivid example for the co-evolution of virus and host.

## Methods

**Expression and purification of HKU2 and SADS-CoV spike ectodomains.** The cDNAs encoding HKU2 spike (YP_001552236) and SADS-CoV spike (AVM41569.1) were synthesized with codons optimized for insect cell expression. HKU2 ectodomains (1–1066) and SADS-CoV ectodomains (1–1068) were separately cloned into pFastBac-Dual vector (Invitrogen) with C-terminal foldon tag for trimerization and Strep tag for purification in the frame of pH promoter. They were expressed in Hi5 insect cell using Bac-to-Bac baculovirus system (Invitrogen). Briefly, the construct was transformed into DH10Bac competent cells and the extracted bacmid was transfected into Sf9 cell using Cellfectin II reagent (Invitrogen). The baculoviruses were harvested after 7–9 days. The high-titer viruses were generated after one more amplification that used to infect Hi5 cells at a density of $1.5 \times 10^6$/ml. After 60 h in culture, the cell medium containing spike ectodomains were concentrated and exchanged to binding buffer (50 mM Tris, pH 8.0, 150 mM NaCl). The spike ectodomains were purified by StrepTactin beads (IBA) and then purified by gel-filtration chromatography using Superose 6 gel filtration column (GE Healthcare) pre-equilibrated with HBS buffer (10 mM HEPES pH 7.2, 150 mM NaCl). Purified ectodomains were concentrated for electron microscopy analysis.

**Cryo-electron microscopy.** Aliquots of spike ectodomains (4 μl, 0.33 mg/ml, in buffer containing 10 mM HEPES pH 7.2, 150 mM NaCl) were applied to glow-discharged holey carbon grids (Quantifoil grid, Au 300 mesh, R1.2/1.3). The grids were then blotted and then plunge-frozen in liquid ethane using FEI Vitrobot system (FEI).

Images were recorded using FEI Titan Krios microscope operating at 300 kV with a Gatan K2 Summit direct electron detector (Gatan Inc.) at Tsinghua University. The automated software (AutoEMation) was used to collect 7663 movies for HKU2 and 4568 movies for SADS-CoV at ×130,000 magnification and at a defocus range between 1 and 3 μm. Each movie has a total accumulate exposure of 49.784 e−/Å² fractionated in 32 frames of 175 ms exposure. Data collection statistics are summarized in Table 1.

Whole frames in each movie were corrected for beam-induced motion using MotionCo2[60]. The final image was bin averaged to give a pixel size of 1.061 Å. The parameters of contrast transfer function (CTF) was estimated for each micrograph using GCTF[61]. Particles were automatically picked using Gautomatch (http://www.mrc-lmb.cam.ac.uk/kzhang/) and extracted using RELION[62]. Initially, ~1,400,000 particles for HKU2 and ~900,000 particles for SADS were subjected to 2D classification. After two or three additional 2D classifications, the best class consisting ~750,000 particles (HKU2) and ~320,000 particles (SADS-CoV) were applied for creating 3D initial model, 3D refinement, and 3D classification. A total of 421,490 particles (HKU2) and 152,334 particles (SADS-CoV) of best classes were selected and subjected to 3D refinement with C3 symmetry to generate density map. The reported resolutions based on the gold-standard FSC cutoff of 0.143 criterion were 2.38 Å for HKU2 spike and 2.83 Å for SADS-CoV spike after RELION post processing. Local resolution variations were estimated using ResMap[63]. Data processing statistics are summarized in Table 1.

**Model building and refinement.** As for the HKU2 spike model building, the initial model of HKU2 S1 NTD was generated by the SWISS-MODEL[64] and fit into the map using UCSF Chimera[65]. The other part of HKU2 spike ectodomains was obtained using Map to Model in PHENIX suit[66]. As for the SADS-CoV spike model building, the initial model was obtained by fit HKU2 S2, NTD, and CTD separately into the map using UCSF Chimera[65]. Manual model rebuilding was carried out using Coot[67] and refined with PHENIX real-space refinement[66]. The quality of the final model was analyzed with Molprobity[68] and EMRinger[69]. The validation statistics of the structural models are summarized in Table 1.

**Reporting summary.** Further information on research design is available in the Nature Research Reporting Summary linked to this article.

## Data availability

The atomic coordinates of HKU2 spike and SADS-CoV spike have been deposited in the Worldwide Protein Data Bank with the accession codes 6M15 and 6M16, respectively; the corresponding maps have been deposited in the Electron Microscopy Data Bank with the accession codes EMD-30037 and EMD-30038, respectively.

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

## Acknowledgements

We thank the Tsinghua University Branch of China National Center for Protein Sciences (Beijing) for the cryo-EM facility and the computational facility support, and L. Zhao, X. Li, J. Wen, H. Zhou, Y. Wang, A. Jia, and S. Zhang for technical support. This work was supported by funds from Beijing Advanced Innovation Center for Structural Biology at Tsinghua University and the National Key Plan for Scientific Research and Development of China (grant number 2016YFD0500307).

## Author contributions

J.Y. carried out protein expression, purification, electron microscopy sample preparation, data collection, image processing, and model building with the help of S.Q. Author X.W. conceived, designed, and directed the study. X.W., J.Y., and R.G analyzed the structure, made the figures, and wrote the manuscript.

## Competing interests

The authors declare no competing interests.
