## [Peer Review File · Nature Communications]

Reviewers' Comments:

Reviewer #1:

Remarks to the Author:

The manuscript from Yu et al reports the cryo-EM structures of the spike protein ectodomains from the bat alphacoronavirus HKU2 and the porcine alphacoronavirus SADS-CoV, which was isolated in 2017 and was responsible for severe watery diarrhea of suckling piglets with a mortality rate of up to 90%. The 3D reconstruction of the HKU2 spike has a claimed resolution of 2.38 Å, which would be the highest resolution coronavirus spike reported to date. Overall, these spikes have some features that are shared with alphacoronavirus spikes, whereas other features are more similar to betacoronavirus spikes. The analysis of the NTDs and clustering into two subtypes was interesting and thoughtful.

Overall, the manuscript is fairly well written, but there are some grammatical issues that need to be addressed. The results are a very thorough analysis of these two new structures and all other coronavirus structures reported to date, but this can be difficult to read at some points due to the variety of acronyms and different nomenclature systems used. The manuscript would likely be better suited to a more specialized journal with a strong structural biology focus. Given that there are more than 10 coronavirus spike structures already reported, the results presented here do not represent a substantial advance in the field, despite the very thorough and thoughtful analysis. There is also some concern noted below about the claimed 2.38 Å resolution.

Major comments

1. The authors claim that the C3 reconstruction for the HKU2 spike has a resolution of 2.38 Å, but in the few figures in which the EM density is shown the resolution appears to be much lower. This is particularly true for supplementary figure 1. At 2.4 Å there should be waters present. Are waters observed in the map? If so, were they added to the model? Some of these concerns could be alleviated by including PDB validation reports for the coordinates and maps, which are now mandatory for submission to numerous journals.

Other comments

1. Line 132 lists a break as spanning 204-204
2. line 158, the phrase 'regarding to the secondary structure feature' can be deleted
3. Line 312 and 313, stand should be replaced with strand

Reviewer #2:

Remarks to the Author:

In the manuscript entitled "Cryo-EM structures of HKU2 and SADS-CoV spike glycoproteins and insights into coronavirus evolution" Yu and co-workers reveal high-resolution cryo-EM structures of bat coronavirus BatCoV-HKU2 and the related swine acute diarrhea coronavirus (SADS-CoV) spike (S) proteins. The authors describe important similarities in the structures of BatCoV-HKU2 and SADS-CoV S proteins and while more compact, they resemble the overall architecture of known CoV S proteins. Having established such overall structural similarities, they performed a detailed comparative analysis of each subdomains of the BatCoV-HKU2 S with that of other coronaviruses for which the structures have already been described. By comparative analysis of the N-terminal domain (NTD) of S the authors show that BatCoV-HKU2 NTD contains a galectin-like β -sandwich fold and is more closely related to the NTD1 (previously named domain 0) domain of the alphacoronavirus HCoV-NL63 than its second NTD domain called NTD2 (previously named domain A). By taking structural differences into account and using comparative analyses with other CoV NTDs, the authors propose to distinguish between two subtypes of NTDs for alphacoronaviruses, with BatCoV-HKU2 NTD being categorized

within NTDs of subtype I along with the NTD1 of HCoV-NL63 and the NTDs of HCoV-229E and NTD2 of HCoV-NL63 falling within subtype II. The C-terminal domain (CTD) of BatCoV-HKU2 was analyzed and Yu and colleagues show that its core adopts a structural fold that is more closely related to that of betacoronavirus CTDs than other alphacoronaviruses. The CTD domain is typically involved in proteinaceous receptor binding and for a number of betacoronaviruses the receptor binding motif (RBM) is known to be located in an extended loop/insert region between β 5 and β 6 strands. However, in the case of BatCoV-HKU2 that loop is short. Taking structural features into account, notably the layers of β -sheets found in CTDs, the authors delineate two major types: the first is named one-layer CTD and corresponds to the CTD of BatCoV-HKU2 and CTDs found in betacoronaviruses such as MHV while the second type is named two-layer CTD and corresponds to CTDs found for other alphacoronaviruses, gamma- and deltacoronaviruses. Comparisons of the SD1 and SD2 regions which connect the CTD with the S2 fusion domain of S, showed that BatCoV-HKU2 retained structural features found in other betacoronavirus S proteins. Interestingly, when authors compared the packing of BatCoV-HKU2 S protomers within a trimer, they found that it adopts an intra-subunit packing where the NTD and CTD of one protomer contact each other. This is also found for other alphacoronavirus S proteins such as those of HCoV-NL63 and HCoV-229E and contrasts with the cross-subunit packing of beta- and gammacoronaviruses. Finally, the S2 fusion domain of BatCoV-HKU2 appears conserved structurally overall, however, due to distinct structural features the S2' cleavage site, located immediately upstream of the fusion peptide is masked by the connecting region (CR) helix and loops.

This is an excellent study which describes for the first time the structure of a bat coronavirus spike protein, BatCoV-HKU2 as well as the S protein of the related and devastating swine virus SADS-CoV. The detailed structural comparisons the authors conducted allowed them to gain broadly significant insights into the molecular evolution of these two viruses, as well as for the evolution of coronavirus S protein structural features, such as the NTD. This work will certainly be very helpful in understanding in more detail the modular nature of the coronavirus spike, and how that fits in with coronavirus evolution and interspecies jumping. Below are a few minor points that would help clarify certain points in the manuscript.

1. The authors have undertaken the task of determining the structures of two related coronavirus S proteins from BatCoV-HKU2 and the swine SADS-CoV. The latter virus is an example of the capacity of coronaviruses to jump from one host species to another, particularly originating from a bat host reservoir. With the ongoing global pandemic of COVID-19, caused by SARS-CoV-2 which likely had origins in a bat species, it would be interesting for authors to put their findings into perspective and discuss the implications of their work in this current context.

2. Line 275: what is the basis for qualifying the β 2 and β 3 strands of SD2 as "primitive"?

3. Lines 422-424: authors state that "The spikes of bat coronavirus HKU2 and porcine coronavirus SADS-CoV (1128 and 1130 residues) are the shortest among all current known coronavirus spike glycoproteins". A paper from 2007 (Dong et al., J Virol 2007, DOI: 10.1128/JVI.00299-07) reported the detection of a divergent coronavirus in Asian leopard cats from Southern China. The S protein of this virus was found to be very short at only 1035 aa, shorter by about 100 aa compared to BatCoV-HKU2 and SADS-CoV (GenBank accession no. ABQ39958.1). Protein alignment analysis shows that the Asian leopard cat S protein is closely related to porcine deltacoronavirus S, but contains a large 125 aa deletion in the NTD. Could authors comment on this unusual coronavirus S protein?

4. Below are a few typos and errors that have been noted:

Line 90: "are essential for the cell entry" consider removing "the" in the sentence

Line 90: "There are also no reports regarding to the", please remove "to" here

Line 227: "To be note", consider rephrasing to "Of note,". This expression is found in other places in

the manuscript (lines 369, 400) and should be changed accordingly.

Line 242: change "stands" with "strands"

Line 283: "HCoV-226E" should be "HCoV-229E"

Line 323: replace "form a turning" with "form a turn"

Line 338: replace "also named as SeACoV" with "also named SeACoV"

Line 406: replace "structural feathers" with "structural features"

Reviewers' comments:

Reviewer #1 (Remarks to the Author):

The manuscript from Yu et al reports the cryo-EM structures of the spike protein ectodomains from the bat alphacoronavirus HKU2 and the porcine alphacoronavirus SADS-CoV, which was isolated in 2017 and was responsible for severe watery diarrhea of suckling piglets with a mortality rate of up to 90%. The 3D reconstruction of the HKU2 spike has a claimed resolution of 2.38 Å, which would be the highest resolution coronavirus spike reported to date. Overall, these spikes have some features that are shared with alphacoronavirus spikes, whereas other features are more similar to betacoronavirus spikes. The analysis of the NTDs and clustering into two subtypes was interesting and thoughtful.

Overall, the manuscript is fairly well written, but there are some grammatical issues that need to be addressed. The results are a very thorough analysis of these two new structures and all other coronavirus structures reported to date, but this can be difficult to read at some points due to the variety of acronyms and different nomenclature systems used. The manuscript would likely be better suited to a more specialized journal with a strong structural biology focus. Given that there are more than 10 coronavirus spike structures already reported, the results presented here do not represent a substantial advance in the field, despite the very thorough and thoughtful analysis. There is also some concern noted below about the claimed 2.38 Å resolution.

Response: We thank the reviewer for the comments and suggestions. The nomenclature of spike domains is not unified yet, two commonly used are: (1) NTD, CTD, SD1 and SD2 of the S1 and the S2; (2) domain 0 and domain A-D of the S1 and the S2. Here we adopted the first nomenclature system, which was used in most recent publications. Our study reported the first side-by-side high resolution structure determinations of the spike from a coronavirus and its homolog with high sequence identity from a bat coronavirus. More importantly, through detailed structural analysis and comparisons, we provided important insights into the structure-function relationship and the evolution of coronavirus spike. One notable example is that the HKU2/SADS-CoV CTD represents a primitive structure in the one-layer CTD family, while the insertion of the extra RBM in β -coronaviruses may represent a recombinant event during evolution. Therefore, we believe that although more than 10 coronavirus spike structures have been determined, our study here still represents an important advance in the field. Also please find our response to the concern about the 2.38 angstrom resolution in the following answer.

Major comments

1. The authors claim that the C3 reconstruction for the HKU2 spike has a resolution of 2.38 Å, but in the few figures in which the EM density is shown the resolution appears to be much lower. This is particularly true for supplementary figure 1. At 2.4 Å there should be waters present. Are waters observed in the map? If so, were they added to the model? Some of these concerns could be alleviated by including PDB validation reports for the coordinates and maps, which are now mandatory for submission to numerous journals.

Response: We thank the reviewer for the suggestions. The reported resolution of HKU2 spike after

post processing in the Relion is 2.38 Å, according to the gold-standard Fourier shell correlation 0.143 criterion (FSC = 0.143). In preparing the previous supplementary Figure 1, we made a mistake by using the unsharpened map directly output from the autorefine module of Relion. In the revised figure, we used the B-sharpened and FOM-weighted map after post processing with the map-sharpening B factor of -87.05 \AA^2 . In the current HKU2 map, the water molecules are clearly visible as exemplified in #Fig. 1, which further supports the 2.38 Å resolution we have reported. As suggested by the reviewer, we added water molecules in the final HKU2 model. The PDB validation reports for the coordinates and maps were also attached with the revised manuscript. In the revised manuscript, we added a new supplementary Figure 3, showing the representative densities of amino acid residues in the HKU2 and SADS-CoV models. Supplementary Figure 5 showed the densities of the CR and FP of the S2 subunit, which are often not well and totally resolved in other coronavirus spike structures.

#Fig.1 Representative densities of water molecules in the HKU2 model

Other comments

1. Line 132 lists a break as spanning 204-204
2. line 158, the phrase 'regarding to the secondary structure feature' can be deleted
3. Line 312 and 313, stand should be replaced with strand

Response: We thank the reviewer for the careful corrections. We have updated the manuscript accordingly.

Reviewer #2 (Remarks to the Author):

In the manuscript entitled "Cryo-EM structures of HKU2 and SADS-CoV spike glycoproteins and insights into coronavirus evolution" Yu and co-workers reveal high-resolution cryo-EM structures of bat coronavirus BatCoV-HKU2 and the related swine acute diarrhea coronavirus (SADS-CoV) spike (S) proteins. The authors describe important similarities in the structures of BatCoV-HKU2 and SADS-CoV S proteins and while more compact, they resemble the overall architecture of known CoV S proteins. Having established such overall structural similarities, they performed a

detailed comparative analysis of each subdomains of the BatCoV-HKU2 S with that of other coronaviruses for which the structures have already been described. By comparative analysis of the N-terminal domain (NTD) of S the authors show that BatCoV-HKU2 NTD contains a galectin-like β -sandwich fold and is more closely related to the NTD1 (previously named domain 0) domain of the alphacoronavirus HCoV-NL63 than its second NTD domain called NTD2 (previously named domain A). By taking structural differences into account and using comparative analyses with other CoV NTDs, the authors propose to distinguish between two subtypes of NTDs for alphacoronaviruses, with BatCoV-HKU2 NTD being categorized within NTDs of subtype I along with the NTD1 of HCoV-NL63 and the NTDs of HCoV-229E and NTD2 of HCoV-NL63 falling within subtype II. The C-terminal domain (CTD) of BatCoV-HKU2 was analyzed and Yu and colleagues show that its core adopts a structural fold that is more closely related to that of betacoronavirus CTDs than other alphacoronaviruses. The CTD domain is typically involved in proteinaceous receptor binding and for a number of betacoronaviruses the receptor binding motif (RBM) is known to be located in an extended loop/insert region between $\beta 5$ and $\beta 6$ strands. However, In the case of BatCoV-HKU2 that loop is short. Taking structural features into account, notably the layers of β -sheets found in CTDs, the authors delineate two major types: the first is named one-layer CTD and corresponds to the CTD of BatCoV-HKU2 and CTDs found in betacoronaviruses such as MHV; while the second type is named two-layer CTD and corresponds to CTDs found for other alphacoronaviruses, gamma- and deltacoronaviruses. Comparisons of the SD1 and SD2 regions which connect the CTD with the S2 fusion domain of S, showed that BatCoV-HKU2 retained structural features found in other betacoronavirus S proteins. Interestingly, when authors compared the packing of BatCoV-HKU2 S protomers within a trimer, they found that it adopts an intra-subunit packing where the NTD and CTD of one protomer contact each other. This is also found for other alphacoronavirus S proteins such as those of HCoV-NL63 and HCoV-229E and contrasts with the cross-subunit packing of beta- and gammacoronaviruses. Finally, the S2 fusion domain of BatCoV-HKU2 appears conserved structurally overall, however, due to distinct structural features the S2' cleavage site, located immediately upstream of the fusion peptide is masked by the connecting region (CR) helix and loops.

This is an excellent study which describes for the first time the structure of a bat coronavirus spike protein, BatCoV-HKU2 as well as the S protein of the related and devastating swine virus SADS-CoV. The detailed structural comparisons the authors conducted allowed them to gain broadly significant insights into the molecular evolution of these two viruses, as well as for the evolution of coronavirus S protein structural features, such as the NTD. This work will certainly be very helpful in understanding in more detail the modular nature of the coronavirus spike, and how that fits in with coronavirus evolution and interspecies jumping. Below are a few minor points that would help clarify certain points in the manuscript.

Response: We really appreciate the reviewer for the encouragements and comments on our study.

1. The authors have undertaken the task of determining the structures of two related coronavirus S proteins from BatCoV-HKU2 and the swine SADS-CoV. The latter virus is an example of the capacity of coronaviruses to jump from one host species to another, particularly originating from

a bat host reservoir. With the ongoing global pandemic of COVID-19, caused by SARS-CoV-2 which likely had origins in a bat species, it would be interesting for authors to put their findings into perspective and discuss the implications of their work in this current context.

Response: Thanks for the very helpful suggestion. We have added some discussion (lines 416-424) in the revised manuscript and a supplementary figure (Supplementary Figure 9) to compare the structure of HKU2 and SARS-CoV-2 S proteins. The information of SARS-CoV-2 was also added in the introduction.

2. Line 275: what is the basis for qualifying the $\beta 2$ and $\beta 3$ strands of SD2 as “primitive”?

Response: As shown in the fig.4b, SD2 domains of HKU2 and β -coronavirus have an additional helix in comparison to those of other species, whereas the remaining structures including the $\beta 2$ and $\beta 3$ strands are conserved among all species. We then suggested that additional helix appears to be an insertion between the $\beta 2$ and $\beta 3$ strands of the SD2, and is retained during evolution of β -coronaviruses. To avoid misunderstanding, we deleted the “primitive” word in the revised manuscript.

3. Lines 422-424: authors state that “The spikes of bat coronavirus HKU2 and porcine coronavirus SADS-CoV (1128 and 1130 residues) are the shortest among all current known coronavirus spike glycoproteins”. A paper from 2007 (Dong et al., J Virol 2007, DOI: 10.1128/JVI.00299-07) reported the detection of a divergent coronavirus in Asian leopard cats from Southern China. The S protein of this virus was found to be very short at only 1035 aa, shorter by about 100 aa compared to BatCoV-HKU2 and SADS-CoV (GenBank accession no. ABQ39958.1). Protein alignment analysis shows that the Asian leopard cat S protein is closely related to porcine deltacoronavirus S, but contains a large 125 aa deletion in the NTD. Could authors comment on this unusual coronavirus S protein?

Response: The Asian leopard cat S protein, termed ALC S in the following, has a truncation in the N-terminus compared to PdCoV S protein, while the rest of the sequence are almost the same. No structure of ALC S is available, so we modeled its structure using SWISS server. As shown in #Fig.2, one strand in the top layer, one strand in the middle layer and one strand in the bottom layer are missing in the ALC S, which could be an adapted feature in the cat coronavirus. The truncation of ALC S NTD could be another adaptation strategy rather than the recruitment of extra domain in the CTD. Since the structure of ALC S is not experimentally determined, we did not include the above discussions into the revised manuscript. However, in order to be more precise, we changed the sentence to “The spikes of SADS-CoV (1130 amino acid residues) and HKU2 (1128 amino acid residues) are among the shortest in the coronavirus spike glycoproteins”. Here we also added the reference for the Asian leopard cat S protein.

#Fig.2 Structural alignment of PdCoV NTD and modeled ALC NTD

4. Below are a few typos and errors that have been noted:

Line 90: “are essential for the cell entry” consider removing “the” in the sentence

Line 90: “There are also no reports regarding to the”, please remove “to” here

Line 227: “To be note”, consider rephrasing to “Of note,”. This expression is found in other places in the manuscript (lines 369, 400) and should be changed accordingly.

Line 242: change “stands” with “strands”

Line 283: “HCoV-226E” should be “HCoV-229E”

Line 323: replace “form a turning” with “form a turn”

Line 338: replace “also named as SeACoV” with “also named SeACoV”

Line 406: replace “structural feathers” with “structural features”

Response: We thank the reviewer for the careful corrections, and we have updated the manuscript accordingly.

Reviewers' Comments:

Reviewer #1:

Remarks to the Author:

All of my concerns have been adequately addressed by the authors.

Reviewer #2:

Remarks to the Author:

The authors have appropriately and carefully addressed all comments and suggestions. The comment and model they provided for asian leopard cat S in their rebuttal letter is greatly appreciated.

Reviewers' comments:

Reviewer #1 (Remarks to the Author):

All of my concerns have been adequately addressed by the authors.

Response: Thank you for the approval.

Reviewer #2 (Remarks to the Author):

The authors have appropriately and carefully addressed all comments and suggestions. The comment and model they provided for asian leopard cat S in their rebuttal letter is greatly appreciated.

Response: Thank you for the comments.